# Sprint performance and force application of tennis players during manual wheelchair propulsion with and without holding a tennis racket

Ilona Alberca[1], Félix Chénier[2,3,4], Marjolaine Astier[1,5], Éric Watelain[1]
Jean-Marc Vallier[1], Didier Pradon[6], Arnaud Faupin[1]*

1 IAPS, Université de Toulon, Toulon, France, 2 Centre for Interdisciplinary Research in Rehabilitation of Greater Montreal, Institut Universitaire sur la Réadaptation en Déficience Physique de Montréal, Montreal, Québec, Canada, 3 Department of Physical Activity Sciences, Université du Québec à Montréal, Montreal, Québec, Canada, 4 Department of Systems Engineering, École de Technologie Supérieure, Montreal, Québec, Canada, 5 Université de Toulon, LAMHESS, Toulon, France, 6 Endicap U1179, UVSQ, Laboratoire d'analyse du Mouvement, Versailles, France

* ilona.alberca@univ-tln.fr

**Data Availability Statement:** Anonymized dataset is available on the public repository Data Archiving

## Abstract

The objective of this exploratory research is to study the impact of holding a tennis racket while propelling a wheelchair on kinetic and temporal parameters in a field-based environment. 13 experienced wheelchair tennis players with disabilities (36.1 ± 8.2 years, 76.8 ± 15.3 kg, 174.8 ± 17.1 cm) classified between 30/8 and first series performed two 20 m sprints in a straight line, on a tennis court: one while holding a tennis racket and the second without a tennis racket. They used their own sports wheelchair. Potential participants were excluded if they had injuries or pain that impaired propulsion. Maximal total force, maximal propulsive moment, rate of rise, maximal power output, push and cycle times and maximal velocity were measured. Sprinting while holding a tennis racket increased the cycle time by 0,051 s and push time by 0,011s. Sprinting while holding a tennis racket decreased the maximal propulsive moment, maximal power output, rate of rise and maximal velocity during propulsion by 6.713 N/m, 151.108 W, 672.500 N/s and 0.429 m/s, respectively. Our results suggest that the biomechanical changes observed associated with racket propulsion are generally in a direction that would be beneficial for the risk of injury. But sprinting holding a racket seems to decrease players propulsion performance. Working on forward accelerations with a tennis racket would be a line of work for coaches.

## Introduction

Wheelchair Tennis (WT) was created in 1970 and included in the Paralympic program in 1992 at the Barcelona Paralympic Games [1, 2]. This sport has gained in popularity throughout the world because it positively affects amputees and persons with spinal cord injuries [3–7]. The trajectory of the tennis ball involves the making of intermittent, multidirectional and non-

and Networked Services (DANS; https://doi.org/10.
17026/dans-xjf-bs8v).

**Funding:** The author(s) received no specific
funding for this work.

**Competing interests:** The authors have declared
that no competing interests exist.

random court movements [7]. Those are subject to intense back and forth movements across
the court such as significant sprints and accelerations [8, 9]. Therefore, the ability to sprint
from a static position prevails in this sport [10]. WT players propel the manual wheelchair
(MW) while holding the racket in their hand, which constitutes the main feature of this sport
which impacts in many ways the propelling of the MW.

Several research studies investigated on the impact of holding a tennis racket on MW propulsion [8, 9, 11], showing a reduction in athlete performance. Indeed, Sindall et al. [9] demonstrated that the movement restrictions created by the holding of a tennis racket had a
significant impact on the player's ability to anticipate the trajectory of the ball, to position himself correctly for striking the ball and to coordinate his propelling movements. In a study with
8 high-level WT players performing sprints while holding and without a tennis racket,
Goosey-Tolfrey and Moss [8] showed that holding a racket interferes with the grip of the handrim and makes pushing ineffective. The maximal velocity on the first three pushes is reduced
by 5.3%, which results in a reduction in the distance covered [8]. De Groot et al. [11], in their
study on sprinting tests on a wheelchair ergometer, observed more power loss before and after
the push while holding a racket due to a reduced push time, which induces inefficiency of
propulsion.

The risk of injury is very present in wheelchair tennis, exceeding even wheelchair basketball
in the percentage of injured athletes. According to a 2012 epidemiological study, 17.9% of
wheelchair tennis players have injuries [12]. Several recurrent injuries exist in this sport: limb
injuries at 72% (tendinopathies, bursitis, ligament, and muscle tension), soft tissue injuries at
30% (blisters, lacerations, abrasions) [13]. However, only one study focused on the impact of
the holding of the racket on the risk of injury [11]. In this study, authors compared the right
hand holding the racket and the left hand, free, during sprint tests on a wheelchair ergometer
[11]. They showed that the holding of the racket induces ineffectiveness in propulsion technique, leading to higher power losses and subsequently higher peak power outputs on the
hand holding the racket during the shortest push phase compared to the free hand. The arm
that holds the tennis racket must support higher forces during wheelchair propulsion in
sprints, compared to the arm without a racket, which is also a criterion of musculoskeletal disorders [14, 15].

The study of the performance and the risk of injury turns out to be important in the case of
WT. Although this two areas are related to numerous temporal and kinetic parameters such as
the velocity, force rate of rise, power output, total force, propulsion moments, etc. [8, 11, 14–
16], only the velocity, push and cycle time, power output and duration of the sprint were measured in the literature according to our knowledge [8, 11]. The analysis of these variables aforementioned could be beneficial for WT. That's why this exploratory research aims to evaluate
the impact of holding a tennis racket on kinetic and temporal parameters in a field-based environment. Specifically, we would like to analyze the impact of the tennis racket during MW
propulsion on maximal total force, maximal propulsive moment, rate of rise, maximal power
output, push and cycle time and maximal velocity. Our exploratory research hypothesis is that
MW propulsion while holding a racket affects the kinetic and temporal parameters in a direction that is coherent with a decreased propulsion efficiency and performance and, an increased
risk of injury.

## Materials and methods

### Study design

Wheelchair tennis players performed two 20 m sprints in a straight line in randomized order:
one while holding a tennis racket (WR) and the second without a tennis racket (WOR). The test

was performed on the tennis courts of the Antibes club during the tournament. The 20 m sprint test is commonly used by athletes in wheelchairs court sports to test their performance [8, 17–19]. Moreover, this test corresponds to the internal logic of WT since it's an intermittent sport with a lot of acceleration phases. They held the racket with their dominant hand which was the right hand for all sportsmen. A rest time of 1 min between the 2 sprints was imposed. Biomechanical parameters were recorded for all sprints. The experimental protocol was approved by the Comité d'Ethique pour les Recherches Non Interventionnelles (CERNI) from Pôle Grenoble Cognition to France (certificate #CERNI 2014-02-07-36 obtained on February 7, 2014).

The participants used their own tennis MW. Their tennis MW had wheel sizes ranging from 24 to 26 inches and a minimum wheel camber of 18˚. Each MW was equipped with one instrumented wheel of 24 or 26 inches on their dominant side (SMARTWheel. 2013 edition, Outfront LCC). Wheel size was chosen depending on the personal tennis MW size of the participants. Those instrumented wheel have a standard weight about 4.7 kg for each wheel size [20]. In accordance with Mason et al. [21], a weight of 2 kg was added to the hub of the second wheel of the MW to compensate for the additional weight and moment of inertia of the instrumented wheel. These wireless instrumented wheels measure the wheel angle and 3D pushrim kinetics at 240 Hz [22–24].

Thanks to the instrumented wheels, the parameters of Table 1 were calculated and averaged on 5 consecutive pushes immediately following reach of maximal velocity, identified from the average wheelchair velocity curve calculated by the SMARTWheel. The pushes were segmented manually based on the minima of the propelling moment (Mz) as shown in Fig 1.

All data processing and calculations were performed using Matlab and the Kinetics Toolkit library created by Félix Chénier (2016). Anonymized dataset is available on the public repository Data Archiving and Networked Services (DANS; https://doi.org/10.17026/dans-xjf-bs8v).

## Participants

13 volunteers WT athletes (36.1 ± 8.2 years. 76.8 ± 15.3 kg. 174.8 ± 17.1 cm) took part in our exploratory study. They were recruited during the French WT tournament of the Antibes club

**Table 1. Description and equations for the outcome measures.**

| Parameters | Description | Equations |
|---|---|---|
| **Pushrim kinetics** | | |
| Maximal totale force ($Ftot_{peak}$) [N] | Sum of the maximum forces in the 3 planes of space applied to the handrim for each push | $sum\left(Fx_{peak}^2 + Fy_{peak}^2 + Fz_{peak}^2\right)^{0.5}$ (2) |
| Maximal propulsive moment ($Mz_{peak}$) [Nm] | Maximal propelling moment applied to the handrim for each push | $peak(Mz)$ (3) |
| Rate of rise (RoR) | Rate of rise in maximal total force for each push | $average(dFtot/dt)$ (4) |
| Maximal power output ($PO_{max}$) [W] | Maximal power output develops by the participant to the handrim for each push | $peak\left[\left(\frac{d\theta}{dt}\right) \times Mz\right]$ (5) |
| **Temporal parameters** | | |
| Push time (PT) [s] | Coupling time between the hand and the handrim for each push | $t_{end}(i) - t_{start}(i)$ (6) |
| Cycle time (CT) [s] | Time between the start of first push and next push for each push | $t_{start2}(i) - t_{start1}(i)$ (7) |
| Maximal velocity ($V_{max}$) [m.s$^{-1}$] | Maximal velocity of the participant | Calculates carried out by the SmartWheel software |

With Fx: horizontal force, Fy: vertical force, Fz: mediolateral force, r: wheel radius, start: start of a push, end: end of a push, t: time (s), θ: wheel angle.

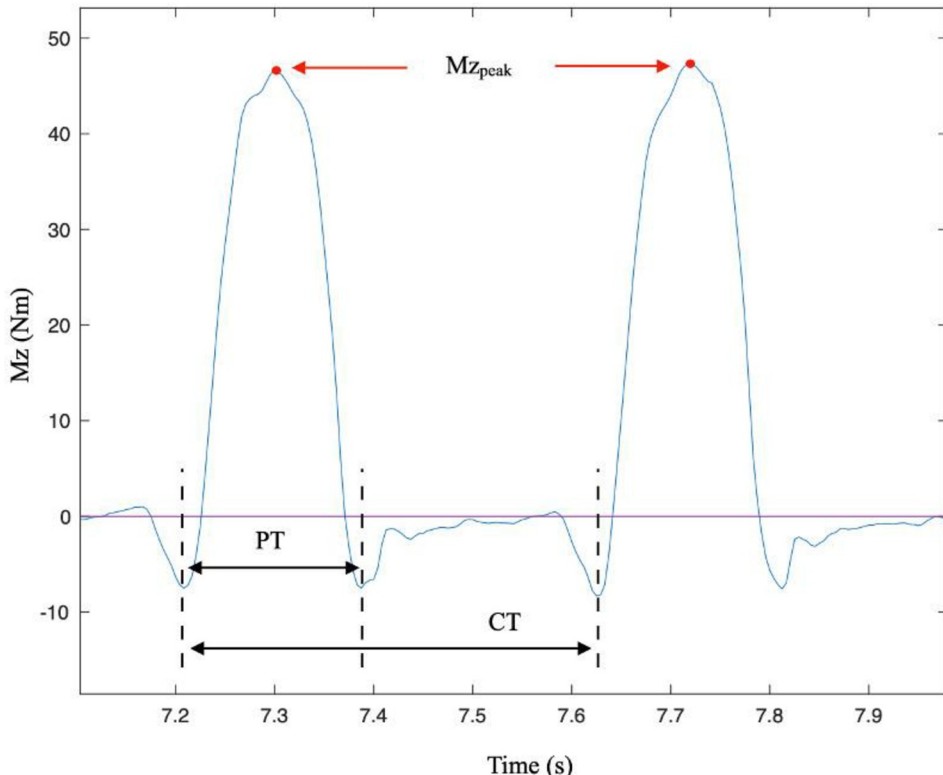

**Fig 1. Manual push segmentation based on the propulsion moments Mz.** With $Mz_{peak}$: maximal propulsive moment, PT: push time, CT: cycle time.

(France) from May 31, 2016 to June 5, 2016. Written informed consents were signed and collected for all participants. They all had a good level of tennis and for some a high level. They were included in the study if they were WT players playing in competitions, over 18 years old and had a minimum ranking of 30/8. This classification in WT corresponds to the results of matches made throughout the sporting year. There are 4 distinct categories: unclassified people, 3rd series people (from 30/6 to 15/7), second series people (from 15/8 to -30/8) and, first series people (national rankings). Regardless of the classification, each player belongs to a division among the following three: man, woman, quad which corresponds to a mixed division. Potential participants were excluded if they had injuries or pain that impaired propulsion. The anthropometric characteristics of the participants, who were all right-handed men, are given in Table 2.

## Ethics

All the participants were informed of the possible risks linked to the experiment and gave their informed consent to participate in the experiment, which took place in accordance with the recommendations of the 1975 Helsinki declaration on the experimentation on human subjects.

## Statistical analysis

The first statistical analysis carried out concerns the statistical power test to determine the sample size necessary for the study. The article by de Groot et al. [25] was used as a reference for this test. Thus, the calculation of the statistical power gave us an average of 8 participants for

**Table 2. Individual anthropometric characteristics of the participants.**

| Subjects | Body mass (kg) | Age (years) | Height (cm) | Disability | Years of practice | Practice level* | Wheel size (inch) |
|---|---|---|---|---|---|---|---|
| 1 | 73.5 | 31 | 169 | Acquired amputation | 1 | 30/8 | 26 |
| 2 | 75 | 23 | 183 | Acquired paraparesis | 3 | 15/8 | 26 |
| 3 | 62 | 32 | 186 | Acquired paraparesis in D12/L1 | 5 | 15/8 | 26 |
| 4 | 46 | 37 | 123 | Acquired amputation | 12 | First series | 24 |
| 5 | 76 | 45 | 171 | Acquired paraplegia in D5 | 25 | 2/8 | 24 |
| 6 | 115 | 40 | 194 | Acquired paraplegia in L1/D12 | 10 | -30/8 | 26 |
| 7 | 79 | 45 | 175 | Paraparesis D12/L1 | 10 | First series | 26 |
| 8 | 79 | 27 | 183 | Left leg acquired polio | 16 | First series | 26 |
| 9 | 70 | 43 | 183 | Acquired paraplegia D12 | 5 | -30/8 | 24 |
| 10 | 78 | 27 | 174 | Muscular atrophy left leg | 8 | 15/8 | 26 |
| 11 | 76 | 32 | 184 | Paraparesis D12/L1 | 7 | First series | 26 |
| 12 | 84 | 50 | 176 | Ankle/left foot arthrodesis +left elbow prosthesis | 8 | 15/8 | 26 |
| 13 | 85 | 37 | 172 | Paraplegia D12 | 8 | 0/8 | 24 |
| Mean ± SD | 76.8 (±15.3) | 36.1 (±8.2) | 174.8 (±17.1) | | 9.1 (±6.2) | | |

With SD: standard deviation, D: dorsal vertebrae, L: lumbar vertebrae.

*: rankings defined by the federation (https://www.fft.fr/competition/paratennis/le-classement-tennis-fauteuil).

the statistical tests that we wanted carried out on our measurements. Based on this average, 13 participants were included in the study. The statistical power was calculated using the G*Power software (G*Power, 2020; g-power.apponic.com).

Each variable was averaged over the participant's five selected pushes and was reported as mean and standard deviations (SD) across all participants. A total of 7 variables were calculated. The aim of our exploratory study is to analyze the impact of using a tennis racket while propelling a tennis MW. To do this, the parameters presented in Table 2 will be compared according to the award condition: without racket and with racket. Precisely, we compare the same dominant hand propelling with and without racquet. First of all a multivariate test was carried out in order to verify the existence of significant differences between the variables measured without and with racket. Once this test has been carried out and verified, data were compared between both conditions (with and without holding a tennis racket) using a paired t-test. Normality was tested for each participant if deviance was found, we used a non-parametric Wilcoxon test. Only the maximal propulsive moment did not check for normality assumptions and thus, a Wilcoxon test was made on this parameter. For each significant difference, the effect size d was calculated using (Eq 1):

$$\text{Eq 1:} \quad d = \frac{mean_{DP} - mean_{DP}}{sd_{CP}} \tag{8}$$

The effect size was interpreted according to Cohen et al. [26]: small (d = 0.2), moderate (d = 0.5) and large (d = 0.8) [26]. Statistics were computed using JASP software (JASP Team, 2016; jasp-stats.org).

## Results

Results of the different kinetic and temporal parameters measured in WOR and WR condition are exposed in Table 3.

Regarding the kinetics parameters, propelling while holding a tennis racket decreased maximal propelling moment ($Mz_{peak}$) ($p = 0.012$), rate of rise (RoR) ($p = 0.033$) and maximal power

**Table 3. Means ± standard deviation of the kinetic and spatiotemporal parameters measured.**

| Parameters | WOR | WR | Differences (WR—WOR) | P-value | d |
|---|---|---|---|---|---|
| **Kinetics** | | | | | |
| Maximal totale force ($Ftot_{peak}$) [N] | 195,387 (±50.11) | 196.531 (±71.790) | -0.429 (±0.809) | 0.686 **NS** | 0.120 |
| Maximal propulsive moment ($Mz_{peak}$) [N.m$^{-1}$] | 39.823 (±8.417) | 33.112 (±10.644) | -6.713 (±4.547) | 0.012* | 0.795 |
| Rate of rise (RoR) [N.s$^{-1}$] | 2988.254 (±1119.702) | 2315.754 (±1064.331) | - 672.500 (±475.529) | 0.033* | 0.701 |
| Maximal power output ($PO_{max}$) [W] | 448.508 (±100.597) | 297.400 (±86.287) | - 151.108 (±106.450) | < 0.001*** | 1.665 |
| **Temporals** | | | | | |
| Push time (PT) [s] | 0.183 (±0.028) | 0.194 (±0.033) | -0.011 (±0.008) | 0.040 * | 0.670 |
| Cycle time (CT) [s] | 0.442 (±0.053) | 0.492 (±0.072) | 0.051 (±0.036) | 0.001** | 1.265 |
| Maximal velocity ($V_{max}$) [m.s$^{-1}$] | 3.527 (±0.343) | 3.098 (±0.368) | -0.429 (±0.303) | < 0.001*** | 1.432 |

With NS: Not significant,

*: p < 0.05,

**: p < 0.01,

***: p < 0.001,

WOR: without racket, WR: with racket.

output ($PO_{peak}$) ($p < 0.001$) but not $Ftot_{peak}$. The differences obtained are classified as moderate to large according to the calculated effect size. Concerning the temporal parameters, propelling while holding a tennis racket increased push time (PT) ($p = 0.04$), cycle time (CT) ($p = 0.001$) and decreased maximal velocity ($V_{peak}$) ($p < 0.001$). The differences obtained are classified as moderate to large according to the calculated effect size.

## Discussion

The aim of our exploratory study was to analyze the impact of wheelchair propulsion while holding a tennis racket on kinetics and temporal parameters in field-based environment. We hypothesized that holding a tennis racket affected these parameters in a direction that is coherent with a decrease in propulsion efficiency and performance, and with an increased risk of injury. This hypothesis has been partially verified. Indeed, maximal propulsive moment, maximal power output and maximal velocity, are positively linked to propulsion efficiency and performance, and they decreased with a moderate to large effect size while holding a tennis racket, which confirms our hypothesis. However, propelling while holding a racket decreased rate of rise and maximal power output and increased cycle time which contradicts our hypothesis since it's considered as protective factors against the risk of injury [15]. Those results suggest that while holding a racket, the biomechanical changes observed are generally in a direction that would be beneficial for the risk of injury, but reduce performance compared to propelling without using the racket.

In our experimentation, we noted a moderate decrease ($d = 0.704$) in maximal propulsive moment by 6.713 N.m$^{-1}$ (±4.547). A decrease in this parameter is associated with a reduction in performance since it corresponds to the moment of force propelling the wheelchair. This result can be explained by the fact that the racket prevents the full grabbing of the handrim during propulsion and thus prevents the athlete from effectively applying the forces and the power used for propulsion. Furthermore, a decrease in the maximal propulsive moment was noted while maximal total force showed no significant difference. When propelling while holding a racket, the participants may have to press harder on the handrim to increase friction and improve the coupling between the hand and the handrim: total force is redirected on the radial and mediolateral axes of the wheels and is useless for propulsion. Moreover, the decline

in maximal propulsive moment is associated with an increase in cycle time while propelling with a racket for the same distance. This means that the athletes appear to produce less and less often effective force and maximal propulsive moment while propelling holding a racket.

The forces used for propulsion, such as the maximal propulsive moment, are reduced when holding the racket. This has the effect of largely reducing (d = 1.208) maximal velocity by 0.429 m.s$^{-1}$ (±0.303) which decreases propulsion performance. Regarding maximal velocity, Goosey-Tolfrey et Moss [8] highlighted a significant decrease in this parameter during wheelchair propulsion while holding a tennis racket, which is coherent with our results. As the participant apparently is no longer able to propel efficiently, their maximal velocity is reduced.

The combination of a decreased maximal propulsive moment and maximal velocity impacts the maximal power output developed by the athlete when propelling the wheelchair. Indeed, maximal power output decreases largely (d = 1.617) by 151.108W (±106.450) while holding the tennis racket to propel the manual wheelchair. This parameter is an important determinant in peak performance [16]. Therefore, a decrease in this parameter induces a decline in propulsion performance. Moreover, a decrease in maximal power output means that the arm holding the racket must suffer less force, which is a protective factor against risk of injury according to Boninger et al. [15]. Nevertheless, De Groot et al. [11] observed an increase in power output when sprinting while holding a racket compared to sprints without a racket. This result would be in contradiction with those of our study. However, it is important to note that we are not comparing the exact same parameters. De Groot et al. [11] were interested in the comparison of the right hand carrying the racket and the left hand without a racquet while, in our study, we compare the same hand with and without a racket. They also compared the power output developed by the same hand with and without a racquet but found no significant difference. In addition, their maximal power output per pushes value is much higher than ours. These differences between their results and those of our exploratory study can be explained by their protocol. De Groot et al. [11] performed their sprint tests on a wheelchair ergometer while in our study, the sprints were completed on a tennis court. This can increase the rolling resistance and thus increase the force and power output required to propel the wheelchair. In addition, their measurement method was not the same as ours since we used a SmartWheel and they used an ergometer. Their statistical analysis included several parametric tests performed on a small cohort (n = 8) which may potentially exclude significant results. Finally, the athletes used the same wheelchair for each test while our athletes used their own different wheelchairs [11].

We also observed a moderate decrease (d = 0.616) in the rate of rise by 672.5 N.s$^{-1}$ (±475.529) while holding a tennis racket which is related to decreased risk of injury according to Boninger et al. [15]. This decrease is seen without significant changes in maximal total force. This means that athletes produce the same maximal total force but in a longer period, hence the decrease in the rate of rise. We can assume that a time delay in the production of the maximal total force is created during the propulsion while holding a racket. As the racket hinders the grip of the handrim, athletes fail to produce maximal total force as quickly as in conventional propulsion conditions. Fig 2 illustrates this idea.

We noted that push time and cycle time increased respectively by 0.011s (±0.008) and by 0.051 (±0.036). These differences are characterized as moderate for the push time (d = 0.670) to large for the cycle time (d = 1.265) according to the effect size obtained for these parameters. An increase in the cycle time, for the same distance traveled, is associated with a decrease in the push frequency which is a component of the cycle time. A reduction in push frequency is notably associated with a decrease in the risk of injury [15]. Concerning temporal parameters, only one study was interested in comparing the push time and cycle time during repeated sprints with and without holding a tennis racket in the same hand [11]. Our results are similar concerning the pushing time: with a racket, it increases. Conversely, de Groot at al. [11] found

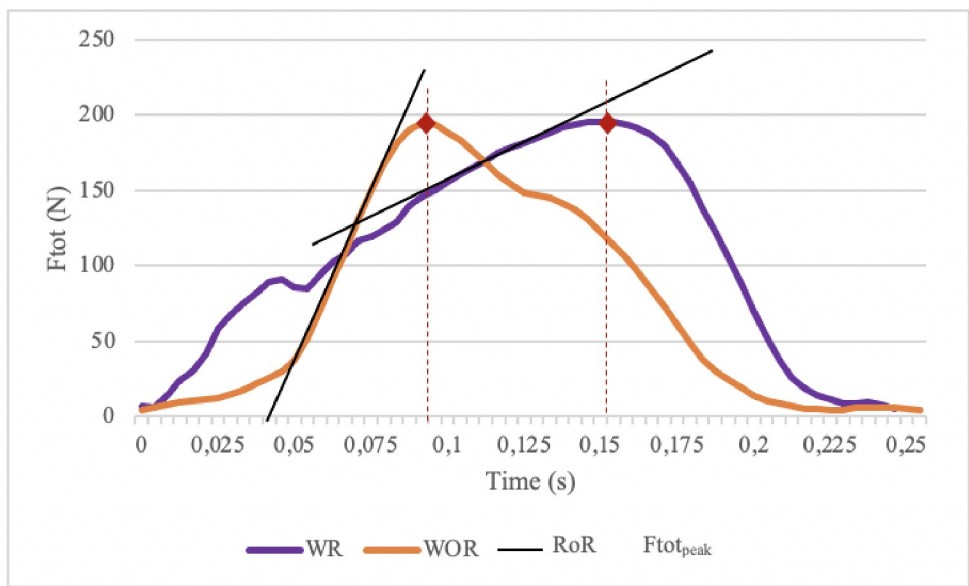

**Fig 2. Comparison of the appearance of the peak of the total force for the subject P5.** With WOR: without racket, WR: with racket, Ror: rate of rise of the total force, $Ftot_{peak}$: maximal total force.

no significant difference for cycle time. We can assume that to compensate for the inefficiency of propulsion while holding a racket, athletes increase their push time to apply the forces on the handrim longer which could explain our results. Moreover, the racket being heavier, the propulsion movement is slower. This leads to an increase in push time and thus, in an increase in cycle time for the same distance traveled.

Regarding the limitation of this study, the first one concerns our population. Indeed, the experiment was carried out on the small sample size, with a large variation in playing experience and disability. But we are confident that this does not affect the general conclusions of the present exploratory study since the measured effect size were moderate to large. In addition, the results of the statistical power test made it possible to determine that a population of 8 athletes was sufficient for our study. Finally, the two reference articles in this field of study share protocols made on populations weaker than ours [8, 25].

Another limitation is the use of a single instrumented wheel. Indeed, we don't know the effect of the racket on the free hand as well as the possible means of compensation developed by this hand to compensate for the ineffectiveness of propulsion while holding a racket. Future studies could explore bilateral propulsion to study the differences between the propelling hand while holding a tennis racket and the free one [11, 27].

It is important to specify that sports wheelchairs have many adjustments that athletes use individually to best adapt their wheelchair to their practice. We can assume that different settings could offer more or less resilience to athletes when using the racquet. The lack of control concerning these settings is a limitation to our exploratory study since they can impact key factors of the athlete's performance such as his stability, rolling resistance or even propulsion efficiency [28]. However, this allows us to respect the ecological conditions of our exploratory study since our experiments were carried out with players' sports wheelchairs and their settings used during tournaments.

Our results seem to show that propelling while holding a racket redirects the force in an unoptimized direction and therefore reduces the propulsion efficiency, which inevitably

impacts athlete's performance. In future studies, it would be interesting to investigate changes in the racket handle to overcome this ineffectiveness, such as modifying the texture and material of the racket handle to increase the coefficient of friction. Koopman et al. [29] put forward this idea, but this solution has not yet been tested.

Finally, our exploratory study revealed significant differences during sprints while holding a racket for certain temporal and kinetic parameters. These results were obtained under maximal test conditions. However, Qi et al. [30] have shown that velocity influences temporal and kinetic parameters. Thus, inducing variations in these parameters for higher velocity [30]. We can assume that our results obtained in maximal tests would be different in submaximal tests. Future studies are needed.

## Conclusion

Our exploratory study looked at the impact of steady-state wheelchair propulsion at maximal velocity while holding a tennis racket on kinetics and temporal parameters related to performance and risk of injury during sprint propulsion on the field. Maximal propulsive moment, maximal power output and maximal velocity decreased in association with an increase in push time and cycle time when propelling while holding a racket. This means that players seem to produce less propulsive moment, which is the force component responsible for propelling the wheelchair, and less often, leading to a decrease in performance. However a decrease in maximal power output, rate of rise and an increase in cycle time are also associated with a reduced risk of musculoskeletique disorders according to Boninger et al. [15]. This suggests that during the steady state sprint propulsion while holding a tennis racket, athletes seem to be less efficient, performant but this kind of propulsion does not appear to add more risk of injury from a temporal and kinetic point of view. Our exploratory study provides details on the training modalities for coaches: the biomechanical changes observed associated with racket propulsion are generally in a direction that would be beneficial for the risk of injury but reduce performance. Working on forward accelerations with a tennis racket would be a line of work for coaches. Since velocity influence temporal and kinetic parameters, it would be interesting to carry out future studies in submaximal condition. Nevertheless, in a context where the performance of wheelchair athletes is increasingly important, it would be interesting to investigate new materials of handgrips and handrim allowing a better match between the hand/tennis racket couple of the user and the handrim of the manual wheelchair.

## Acknowledgments

The authors thank the various participants in the study as well as the laboratory of the IAPS.

## Author Contributions

**Data curation:** Ilona Alberca, Félix Chénier, Didier Pradon.

**Formal analysis:** Ilona Alberca.

**Funding acquisition:** Marjolaine Astier.

**Methodology:** Marjolaine Astier, Arnaud Faupin.

**Project administration:** Jean-Marc Vallier, Arnaud Faupin.

**Validation:** Jean-Marc Vallier.

**Writing – original draft:** Ilona Alberca.

**Writing – review & editing:** Félix Chénier, Éric Watelain, Arnaud Faupin.

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
