## [Decision Letter · Decision Letter 0]

6 Sep 2021

PONE-D-21-16314Sprint performance and force application of tennis players during manual wheelchair propulsion with and without holding a tennis racket.PLOS ONE

Dear Dr. Alberca,

Thank you for submitting your manuscript to PLOS ONE. After careful consideration, we feel that it has merit but does not fully meet PLOS ONE’s publication criteria as it currently stands. Therefore, we invite you to submit a revised version of the manuscript that addresses the points raised during the review process.

We look forward to receiving your revised manuscript.

Kind regards,

Yih-Kuen Jan, PhD, University of Illinois at Urbana-Champaign

Academic Editor

PLOS ONE

2. In the method section of your manuscript, please report 1) the date of ethics approval, and 2) the date range of participants' recruitment.

3.Please include your full ethics statement in the ‘Methods’ section of your manuscript file. In your statement, please include the full name of the IRB or ethics committee who approved or waived your study, as well as whether or not you obtained informed written or verbal consent. If consent was waived for your study, please include this information in your statement as well.

Additional Editor Comments (if provided):

Reviewers' comments:

Reviewer's Responses to Questions

**Comments to the Author**

1. Is the manuscript technically sound, and do the data support the conclusions?

Reviewer #2: Yes

Reviewer #3: Partly

2. Has the statistical analysis been performed appropriately and rigorously? 

Reviewer #2: Yes

Reviewer #3: I Don't Know

3. Have the authors made all data underlying the findings in their manuscript fully available?

Reviewer #2: Yes

Reviewer #3: Yes

4. Is the manuscript presented in an intelligible fashion and written in standard English?

Reviewer #2: Yes

Reviewer #3: Yes

5. Review Comments to the Author

Reviewer #2: This study aimed to study the impact of holding a tennis racket while propelling a wheelchair on kinetic and temporal parameters in a field-based environment.

- The main concern of this study is: the sample size of 13 is very small to provide and generalize the findings.

- The main criteria of the study participants is not clarified in the abstract.

- The design of the study is not demonstrated.

- No clear message or recommendation is demonstrated in the abstract.

- Generally, the abstract needs to be rewritten.

- Introduction: The significance of the study needs more clarification.

- The authors should reframe the methods in accordance with components (SPICES) for methods

1. Study design, setting, sample size

2. Participant

3. Intervention/issue of interest (exposure)

4. Comparison

5. Ethics and endpoint

6. Statistical analysis

- How was sample size determined?

- How and who administrates the data collected?

Reviewer #3: The authors examined differences in hand rim propulsion biomechanics during a steady state sprint between wheelchair tennis athletes with or without a racket in hand. Overall, the study is well written and designed and parallels the structure of previous studies pertaining to the examination of every day propulsion biomechanics. Furthermore, the authors’ general interpretation of findings are reasonable(depending on how they answer comments below). I have only one moderate statistical concern and methodological comments which could be impactful to the overall picture of the study depending on how the authors respond. First, the authors have utilized paired analysis (parametric and non-parametric versions) however it’s unclear if they used a correction for multiple comparison. In essence, they may have conducted numerous independent paired tests (one for each dependent variable ?) which is problematic. If this is the case, they should divide alpha by the # of comparisons (bonferroni correction) or elect to use a multivariate test which has the benefit of auto correction and effect size calculation simultaneously. Finally, the authors state participants used their own manual chairs but didn’t report camber angle or more importantly specify(unless I missed this) if they were tennis chairs or everyday chairs. Tennis chairs typically have large amounts of camber which alters the interface or coupling of the hands with the handrim (hand and wrist angle). This could influence propulsion mechanics adaptability in the presence or absence of a racket. If the study was examined with everyday chairs lacking camber it presents a huge limitation which needs to be discussed. In fact it may be necessary to generalize results only to those using similar set ups. A final limitation would also include lack of control for or reporting of wheelchair configuration elements like rear wheel axle position, seat height, seat drop etc. These metrics are commonly used and exaggerated in tennis configurations which may offer athletes more resilience while holding a racket since they effect propulsion efficiency and trunk balance.

6. PLOS authors have the option to publish the peer review history of their article (what does this mean?). If published, this will include your full peer review and any attached files.

Reviewer #2: **Yes: **Walid Kamal Abdelbasset

Reviewer #3: **Yes: **Ian M Rice

---

## [Author Response · Author response to Decision Letter 0]

4 Jan 2022

Dear reviewers, 

Thank you for your work and your feedback on this article "Sprint performance and force application of tennis players during manual wheelchair propulsion with and without holding a tennis racket”. Here are the answers I was able to give to each of your comments.

Response reviewer 2:

Thank you for your comments, which help to improve the article, particularly on its methodology.

Regarding the main concern of the study, we have a small population. I added description of this limitation in the discussion. In addition, a statistical power test based on the reference article by de Groot et al. (2017) was carried out and determined that 8 subjects are sufficient in our study and in the statistical tests we carried out. This point was enhanced in discussion and methodology.

Based on your comments, I also incorporated several elements into the summary, including participant criteria, a reference for the study design, and a message of recommendations.

I clarified the meaning of the study in the introduction as you advised me.

Finally, concerning the methodology and the SPICES criteria, our article is an exploratory research article. Consequently, the methodology used during the experiments cannot fully correspond to the SPICES components. In order to incorporate this remark, I have nevertheless modified this part and added elements to best respond to your comment. I hope this will suit you.

Response reviewer 3:

Thank you for your very constructive and motivating feedback.

Regarding the statistical part, it has been corrected according to your advice. After consulting several experts, my choice fell on a multivariate test to ensure that there was a significant difference between the parameters measured depending on the test condition. Once this test was validated, I refined the variable-to-variable analysis with a T-test. I think the combination of these two tests strengthens our statistics and best matches the comparison we want to make based on the data we have. I added the information in the statistical analysis. In addition, due to the exploratory nature of this research and the statistics, I modulated the results to be conditional and not affirmative.

Finally, I added some clarifications. Hence the participants used their own manual tennis wheelchairs. I only have the measurement of the size of the wheelchair wheels of each athlete, but without having everyone's camber angle, we know that athletes all had a minimum angle of 18°. I can add this clarification. Otherwise, I wrote a section in limitations regarding the lack of knowledge of chair measurements. While this is a limitation, it is also a strength to our study. Indeed, we evaluated the athletes into their own wheelchairs with the same settings as those used in competition. We are therefore as close as possible to an experiment in ecological condition.

I hope you find these answers and revisions satisfactory. I remain at your disposal to answer any questions you may have.

Your sincerely,

Ilona Alberca

---

## [Decision Letter · Decision Letter 1]

19 Jan 2022

Sprint performance and force application of tennis players during manual wheelchair propulsion with and without holding a tennis racket.

PONE-D-21-16314R1

Dear Dr. Alberca,

We’re pleased to inform you that your manuscript has been judged scientifically suitable for publication and will be formally accepted for publication once it meets all outstanding technical requirements.

Kind regards,

Yih-Kuen Jan, PhD

Academic Editor

PLOS ONE

Additional Editor Comments (optional):

Reviewers' comments:

Reviewer's Responses to Questions

**Comments to the Author**

1. If the authors have adequately addressed your comments raised in a previous round of review and you feel that this manuscript is now acceptable for publication, you may indicate that here to bypass the “Comments to the Author” section, enter your conflict of interest statement in the “Confidential to Editor” section, and submit your "Accept" recommendation.

Reviewer #1: All comments have been addressed

Reviewer #2: All comments have been addressed

2. Is the manuscript technically sound, and do the data support the conclusions?

Reviewer #1: Yes

Reviewer #2: Yes

3. Has the statistical analysis been performed appropriately and rigorously? 

Reviewer #1: Yes

Reviewer #2: Yes

4. Have the authors made all data underlying the findings in their manuscript fully available?

Reviewer #1: Yes

Reviewer #2: Yes

5. Is the manuscript presented in an intelligible fashion and written in standard English?

Reviewer #1: Yes

Reviewer #2: Yes

6. Review Comments to the Author

Reviewer #1: Thanks to the authors,new topic , good article and analysis , the commebts have been addressed correctly

Reviewer #2: All comments have been addressed. No further comments are required. The publication could be published in the current form.

7. PLOS authors have the option to publish the peer review history of their article (what does this mean?). If published, this will include your full peer review and any attached files.

Reviewer #1: No

Reviewer #2: **Yes: **Walid Kamal Abdelbasset

---

## [Editor Report · Acceptance letter]

26 Jan 2022

PONE-D-21-16314R1 

Sprint performance and force application of tennis players during manual wheelchair propulsion with and without holding a tennis racket. 

Dear Dr. Alberca:

I'm pleased to inform you that your manuscript has been deemed suitable for publication in PLOS ONE. Congratulations! Your manuscript is now with our production department. 

Kind regards, 

on behalf of

Dr. Yih-Kuen Jan 

Academic Editor

PLOS ONE